# Implementation of the Department of Housing and Urban Development’s Smoke-Free Rule: A Socio-Ecological Qualitative Assessment of Administrator and Resident Perceptions

**DOI:** 10.3390/ijerph18178908

**Published:** 2021-08-24

**Authors:** Kimberly Horn, Sallie Beth Johnson, Sofía Rincón-Gallardo Patiño, Kevin Krost, Tiffany Gray, Craig Dearfield, Chenguang Du, Debra Bernat

**Affiliations:** 1Department of Population Health Sciences, Virginia Tech Carilion Research Institute, Riverside Circle, Roanoke, VA 24016, USA; 2Department of Public Health and Healthcare Leadership, Radford University Carilion, 101 Elm Avenue, SE, Roanoke, VA 24013, USA; sejohnson1@radford.edu; 3Department of Human Nutrition, Foods, and Exercise, Virginia Tech, Blacksburg, VA 24060, USA; sofiargp@vt.edu; 4Department of Leadership, Counseling, and Research, Virginia Tech, Blacksburg, VA 24060, USA; kevinkrost@vt.edu (K.K.); dcheng6@vt.edu (C.D.); 5Department of Epidemiology, The Milken Institute School of Public Health, The George Washington University, 905 New Hampshire Avenue, Northwest, Washington, DC 20052, USA; tgray85@gwu.edu (T.G.); cdearfield@gwu.edu (C.D.); dbernat@gwu.edu (D.B.)

**Keywords:** smoke-free policy, housing, tobacco control, smoking, health policy, socio-ecological framework, qualitative

## Abstract

In July 2018, the United States Department of Housing and Urban Development (HUD) implemented a mandatory smoke-free rule in public housing. This study assessed administrator and resident perceptions of rule implementation during its initial year in the District of Columbia Housing Authority (DCHA). Assessment included nine focus groups (*n* = 69) with residents and in-depth interviews with administrators (*n* = 7) and residents (*n* = 26) from 14 DCHA communities (family = 7 and senior/disabled = 7). Semi-structured discussion guides based on the multi-level socio-ecological framework captured dialogue that was recorded, transcribed verbatim, and coded inductively. Emerging major themes for each socio-ecological framework level included: (1) Individual: the rule was supported due to perceived health benefits, with stronger support among non-smokers; (2) Interpersonal: limiting secondhand smoke exposure was perceived as a positive for vulnerable residents; (3) Organizational: communication, signage, and cessation support was perceived as a need; (4) Community: residents perceived mobility, disability, weather, and safety-related issues as barriers; and (5) Public Policy: lease amendments were perceived as enablers of rule implementation but expressed confusion about violations and enforcement. A majority of administrators and residents reported favorable implications of the mandated HUD rule. The novel application of a socio-ecological framework, however, detected implementation nuances that required improvements on multiple levels, including more signage, cessation support, clarification of enforcement roles, and addressing safety concerns.

## 1. Introduction

Smoke-free policies in public spaces (i.e., workplaces, restaurants, and bars) show success in reducing secondhand smoke exposure [1], tobacco use, and a host of negative tobacco-related health outcomes [2]. Unlike public spaces, smoking in private homes is largely unregulated because of ethical concerns about regulating individuals’ private behaviors. Residents in multi-unit housing are particularly susceptible to involuntary secondhand smoke exposure [3]. Homes, therefore, remain a primary place of involuntary and significant secondhand smoke exposure among adults and children, where smoke can enter through ventilation ducts, doors, and other sources [3].

The health burden of secondhand smoke exposure in multi-unit housing is underscored by approximately 80 million US citizens living in multi-unit housing. Nearly 7 million of those residents, including 775,000 children, reside in government-subsidized multi-unit housing [4]. Residents of government-subsidized multi-unit housing represent underserved populations who are already at high risk for chronic diseases and overall poorer health [4]. Government-subsidized multi-unit houses have a high percentage of low-income individuals, racial and ethnic minorities, seniors, and children. Residents also have smoking rates and tobacco-related health disparities significantly higher than the general population. Recognizing this problem, the US Department of Housing and Urban Development implemented a mandatory smoke-free rule in 2018, hereafter referred to as the “non-smoking rule”, prohibiting the use of lit tobacco products, including cigarettes, cigars, pipes, and waterpipes (hookah), in all living units, common areas, administrative office buildings, and outdoor areas of up to 25 feet from Housing Authority property. The rule does not explicitly prohibit electronic nicotine delivery systems but leaves that to the discretion of individual Housing Authorities. Specific components of the non-smoking rule are described in detail in Table 1, along with several suggestions for how to implement each of the components. A primary goal of the rule was to create healthier homes for families living in public housing by improving indoor air quality [5].

Purpose of the rule: to improve indoor air quality in the housing; benefits the health of public housing residents, visitors, and public housing authority staff; reduces the risk of catastrophic fires; and lowers overall maintenance costs. Full compliance by 30 July 2018 [5].

### Previous Research on Smoke-Free Policies

While smokers often report the benefits of smoke-free policies, they also reported that these policies are unfair [6]. Studies suggest potential benefits of smoke-free multi-unit housing policies on smoking behavior and health [7], including decreased secondhand smoke exposure [8], reduced cigarette consumption [9], and increased cessation rates [10]. A study of Portland, Oregon residents found that frequent exposure to secondhand smoke declined significantly after the smoke-free multi-unit housing policy was implemented, with 41% reporting frequent exposure prior to implementation and 17% reporting frequent exposure after 16 months [8]. Data showed that residents reported that the smoke-free policy was an important part of their decision to quit or reduce their cigarette consumption. While self-reported compliance with the policy was high, 17% of residents reported smoking in their units after the policy was implemented. Snyder and other researchers highlight the importance of local data collection in evaluations of smoke-free policies for better understanding the effects of smoke-free policies, particularly improving our understanding of resident preferences and perspectives of multi-unit housing operators [11].

When evaluating smoke-free policies it is necessary to examine individual- and group-level data as well as aggregated data. While health outcomes of smoke-free policies in public places have been well documented in the general population, evidence suggests that the positive effects may not equally apply across sex, socio-economic status, race and ethnicity, and age groups and may be important factors to consider [12]. A study by Levy et al. showed that smoke-free laws were associated with lower rates of smoking for females with higher education, while this relationship was not significant for females with lower education levels [13]. Mead et al. found a significant decline in acute myocardial infarction hospitalizations for non-Hispanic white adults following a statewide smoke-free policy, while no changes were observed for other racial or ethnic groups [14]. A study of the effect of the smoke-free law in New York City, using cotinine, found that certain groups, including individuals of low socio-economic status, are still exposed to secondhand smoke despite the comprehensive law [15].

To date, the literature reflects a fragmented approach to understanding the effects of smoke-free policies. The present study applied a social-ecological framework as a novel way to emphasize individual and social environmental factors as necessary targets for evaluation and improvement. Moreover, the framework supports the study assumption that appropriate changes in the multi-unit housing environment will produce changes in individuals and that the support of individuals in the community is essential for implementing effective changes in the home environment [16,17,18]. Accordingly, the purpose of this study is to examine administrator and resident perceptions on multiple dimensions (i.e., individual, interpersonal, organizational, community, and public policy) of the Department of Housing and Urban Development mandated smoke-free rule during its initial year of implementation in District of Columbia Housing Authority multi-unit housing communities.

## 2. Methods

### 2.1. Study Sample

Recruitment focused on DCHA residents from 14 properties, (7 family and 7 senior/disabled). Based on DCHA classification, family properties included low-rise and townhome building types. For senior/disabled residents, their building type tended to be high-rise buildings that were designed differently to accommodate mobility issues, which make them particularly vulnerable populations to secondhand smoke exposure. The team recruited a total convenience sample of 109 residents and 7 administrators from the 14 properties. Sixty-nine residents participated in nine baseline focus groups. Sixty-one residents participated in individual in-depth interviews. Twenty-one residents participated in both focus groups and interviews. The convenience sample was derived from residents who had participated in previous research activities and consented to participate in additional studies. Furthermore, focus group participants could bring friends from their public housing community to participate in the focus group, but this was not our primary source of sampling participants. Eligibility included District of Columbia Housing Authority property residents between that the ages of 18–80 years. The seven administrators volunteered to participate in key stakeholder interviews. All study methods received DCHA community partner review and IRB approval (#180523) from George Washington University.

### 2.2. Data Collection

#### 2.2.1. Demographic Information

Resident demographics were obtained through surveys administered separately from the focus group and individual interviews. Not all participants completed the survey, which led to missing data. There were 21 focus group participants and four in-depth interview participants who had missing demographic information. Any demographic information reported below is based on all the data that was available. Data was collected after implementation of the smoke-free rule, with focus groups between October 2018 and December 2018 and individual interviews between November 2018 and November 2019.

#### 2.2.2. Focus Groups

Focus groups occurred in a community space (e.g., community room) at each property. Separate focus groups were conducted for smokers and non-smokers; the number of participants in each focus group is summarized in Table 2. A focus group moderator’s guide provided scripts for the administration and implementation of the focus groups [19,20]. One team member led each focus group, while another team member assisted and took notes. Participants were provided snacks and, upon completion, were given a $25 gift card for their participation. All focus groups were recorded and professionally transcribed verbatim by Landmark Associates Inc.

#### 2.2.3. Individual Interviews

Individual interviews occurred in a private community space (e.g., community room) at each property or via telephone. Among the 61 resident interviews, 26 were non-smokers and 35 were smokers. Among the seven administrator interviews, six were non-smokers and one was a smoker. A pilot-tested interview protocol was developed to provide the interviewers with a script. The interviews lasted 15–30 min and participants received $25 for their participation upon completion. All interviews were recorded and professionally transcribed verbatim by Landmark Associates Inc. 2.2.4. Interview and Focus Group Moderator’s Guides.

The focus groups and individual interviews assessed reactions to rule implementation, particularly how residents learned about the rule, how much the residents knew about the nature of the rule (e.g., products included, where they could smoke, enforcement), their concerns about the rule, resident involvement in the implementation process, perceived barriers to compliance with the rule, barriers to its equitable implementation across the diverse public housing developments, and perceived benefits of the rule.

The focus group design allowed participants the time and opportunity to reflect on The Department of Housing and Urban Development’s rule, and thus entered into the process of making implicit knowledge explicit. As well, it permitted residents to revisit their own and others’ explanations and clarify, revise, or expand on previous statements that allowed theoretical saturation within the group. The design provided an opportunity for preliminary analyses of the data and construct questions provoked by the group’s discussion about the Department of Housing and Urban Development rule or to address the ambiguity [21].

### 2.3. Data Analysis

Interviews were inductively coded using grounded theory [22] to ensure saturation of codes, the ability to generate theory, and reduction of the influence of preconceptions on the results. Themes were developed from related codes that captured perceptions of the non-smoking rule, it’s implementation in public houses, and barriers and facilitators of the rule. The study incorporated an adapted nominal group method of recording themes and an iterative research process consistent with grounded theory. This method allowed for development of thematic areas and categories fitting with the socio-ecological framework (Figure 1) while using inductive coding, which makes it a unique qualitative research method. The applied social-ecological model allowed the policy effects to be evaluated holistically among residents and administrators at the District of Columbia Housing Authority rather than piecewise, which is often the case. More specifically, as shown in Figure 1, the present investigation examined the multiple spheres or domains of potential relation to the rule implementation and effects [23].

Codes were grouped into concepts and further grouped into categories. Finally, categories evolved into themes following iterative consensus coding. A resulting codebook provided logic and breadth. Using the final codebook structure, two members of the research team separately coded all of the transcripts using Dedoose version 8.3.47b [24] (Socio Cultural Research Consultants (SCRC), Los Angeles, CA, USA), a mixed methods software package used to facilitate data management and organization. Disputed codes were jointly resolved by the research team through consensus meetings for any codes not obtaining an inter-rater reliability >0.80. Triangulation [25] utilizing the interrater reliability functionality of Dedoose selected excerpts with pre-specified codes from various transcripts that each coder independently coded and then compared and discussed the results. Dedoose calculated Cohen’s Kappa coefficient to measure inter-rater reliability [26].

## 3. Results

### 3.1. Participant Demographics

There were seven administrators representing the District of Columbia Housing Authority (from senior/disabled and family communities) who participated in in-depth interviews. Demographic data was not collected on administrators.

Of the total resident participants in the overall study (*n* = 109), 69 participated in 9 focus groups (nonsmoker group = 4, smoker group = 5). The average focus group size was 7.63 participants, with a standard deviation of 2.72. For demographic survey data connected to focus groups and individual interviews, the majority of the respondents were female (85%), smokers (59.4%), and living in family buildings (72.5%). The mean age was 56.37, with a standard deviation of 12.8. Several residents did not respond to all of the demographic questions but were included in the sample wherever possible. Race and ethnicity data were missing from 41% of the sample. The DCHA community is 90% Black/African American.

### 3.2. Overall Study Themes

The main study themes, categories, and definitions for coding are described in Table 3. The study identified seven unique themes aligned with the smoke-free rule components that emerged from the coding of meaning units (i.e., a word, phrase, or quote expressing a distinct idea) [27] from non-exclusive categories: cessation support, *n* = 100 meaning units, communication, *n* = 150 meaning units, enforcement, *n* = 69 meaning units, lease, *n* = 38 meaning units, resident Engagement, *n* = 175 meaning units, signage, *n* = 344 meaning units, and violations, *n* = 568 meaning units. A majority of the meaning units revolved around concerns regarding implementation of the smoke-free rule, specifically related to signage and violations. For each theme, multiple categories evolved for both administrator and residents.

The themes for the study aligned with multiple levels of the socio-ecological model: (1) individual, (2) interpersonal, (3) organizational, (4) community and (5) public policy. A majority of meaning units from the discussions were aligned with themes at the organizational level (*n* = 1199 meaning units). The least discussion occurred with themes aligned at the interpersonal level (*n* = 375 meaning units). At the individual level, there were 791 meaning units in total. At the community level, there were 912 meaning units in total. At the public policy level, there were 606 meaning units in total. Table 4 displays the themes associated with each level along with illustrative quotes presented by administrators, resident smokers, and resident non-smokers.

## 4. Discussion

This study presents results from in-depth interviews and focus groups with administrators and residents revealing multiple factors at different levels of the socio-ecological framework that influences perceptions and uptake of the non-smoking rule in the District of Columbia Housing Authority public housing. These factors occurred at the individual, interpersonal, organizational, community, and public policy level, and included both positive and negative factors. These results both converged and diverged with the results of previous studies on similar topics.

Findings have several implications for future similar programs and interventions. In general, it is important to improve and provide more access to tobacco cessation services to ensure that individuals who want to quit tobacco have the resources to do so. Moreover, there needs to be ample training for administrators and supervisors regarding policy communication, in advance and ongoing. Related to policy communication, both residents and administrators indicated the necessity of more signage about the rule and where individuals can and cannot smoke. There was much uncertainty for both administrators and residents about the policy and its enforcement. More nuanced implications are discussed next by level.

### 4.1. Individual

Generally, the smoke-free rule was supported by administrators and residents due to perceived health benefits. Resident nonsmokers expressed stronger support than resident smokers. The locations included in the smoke-free rule affected the degree of support and implementation success, with outside areas and the 25 feet component of the rule causing the most confusion and dissent for both administrators and residents. Three main themes reflected the individual level: lease, resident engagement, and violations. Many participants noted that addendums to residents’ lease agreements were used to communicate the components of the smoke-free rule, promote compliance, and gain buy-in for implementation. Cessation support services were reported by administrators as helpful for engaging residents with the policy implementation process. However, several resident smokers were not interested in cessation support or quitting smoking. Most participants were aware of the smoke-free rule and reported a variety of consequences for violations, such as verbal and written warnings. Residents preferred to report violations to administrators, rather than confront fellow residents or visitors who were not complying with the rule.

These findings are consistent with prior research showing individual-level rule support, especially among non-smokers [27,28,29]. Past research suggests that most of the support is driven by perceived health benefits of the rule [30]. However, despite the support and perceived benefits, there are consistent compliance barriers or resistance making enforcement a challenge for smoking-free rules no matter the geographic location or setting [8,9,27,30,31,32,33,34]. Studies show resident support for smoke-free multi-unit housing policies [3,8,35,36], including racially and ethnically diverse, low-income seniors [37], and property managers [3,8,38], although important differences existed by smoking status [39].

### 4.2. Interpersonal

The reduction of secondhand smoke exposure was perceived as a positive effect of the smoke-free rule for vulnerable residents. Based on interactions between residents and administrators, it was reported that the time-frame used to communicate the smoke-free rule was not adequate. Findings also pointed to a need for clearer dialogue about who is in charge of the enforcement of the rule and the consequences of violating the policy. Smokers consistently stated unwillingness to report people smoking in undesignated smoking areas, while nonsmokers were slightly more willing to report rule violations. Administrators and residents both mentioned that signing the lease agreement was mandatory and, notably, the primary way several residents were officially notified about the enactment of the smoke-free rule. As other research shows, the present study found that the rule uses the lease as a facilitator for policy implementation [31,34].

Residents expressed concerns about reporting other resident violations to avoid complicated relationships with their neighbors [31]. Residents consistently indicated an active unwillingness to report people smoking to avoid conflicts [31].

### 4.3. Organizational

At the organizational level, many administrators and residents expressed disappointment in the District of Columbia Housing Authority’s actions regarding the smoke-free housing rule. The cessation support provided to residents interested in quitting smoking was reported as inadequate by both administrators and residents, in essence, a broken promise. Both smoking and non-smoking residents indicated a desire for cessation materials to be provided.

Many residents and administrators said that there were few signs on property that indicated that smoking was not allowed on the property or 25 feet from the building, possibly indicating a lack of supply from the District of Columbia Housing Authority. This led to many residents and administrators not being aware of the rule’s implementation and confusion about where smoking on the property was allowed. Related to this, many of the participants believed the rule communication was inadequate, while others thought the amount of communication was acceptable. Several residents mentioned receiving letters about the rule, formal meetings, and word-of-mouth communication, while others said they did not hear any information before the rule was implemented. This lack of overall communication and signage from the District of Columbia Housing Authority affected the enforcement of the rule, with both residents and administrators saying they did not know who was in charge of enforcement. Mostly, this led to a lack of enforcement and increased non-compliance with the rule.

### 4.4. Community

There were several barriers to the rule implementation at the community level based on the environment of family and senior/disabled housing units. Resident mobility was mentioned by many administrators and residents as a barrier to the rule’s implementation, due to the environment of the community. Several residents mentioned disability as a factor that kept themselves or other residents from complying with the rule, due to the difficulty that persons with disabilities might have going 25 feet away from their building. Many residents mentioned the absence of a designated covered area on the property where they could smoke and be compliant. Rain, snow, and heat were all mentioned as weather that made it difficult or undesirable to comply with the rule by going outside. Safety for themselves and others was mentioned several times as a barrier for compliance. Administrators and residents mentioned the prevalence of crime in their neighborhood and concern about smoking alone or with others outside of their property, especially at night or in areas where there have been reports of violence or drugs. Many also discussed having to leave children home alone or unsupervised when they left their building to smoke, or not wanting to go outside so they could watch their children. The unique discussion on safety issues in the community is consistent with several studies that examined the implementation of government-subsidized multi-unit housing, specifically, in urban centers including Portland, Oregon [8] and Boston, Massachusetts [29].

### 4.5. Public Policy

At the public policy level, the most prevalent themes related to the impact of the District of Columbia Housing Authority smoke-free rule were resident lease requirements and rule violations. Many residents and administrators mentioned that they were required to sign an addendum to their lease agreeing to abide by the rule. This helped enable the implementation of the rule because it ensured that the residents were aware of the rule and informed the residents of potential consequences if they did not comply with the rule. Both residents and administrators mentioned marijuana as having an impact, with many thinking that it was included in the smoke-free rule and that they had to be compliant with it as well. There was no mention of residents being evicted as a result of non-compliance, but many residents were aware that they could receive three warnings before they could get evicted.

According to the Department of Housing and Urban Development, “it is important to evaluate various aspects of the implementation of the rule by the public housing authorities, including the benefits on indoor air quality and resident health as well as the actual implementation process” (Change is in the Air, 2014, HUD). Accordingly, there are several strong takeaways about the Department of Housing and Urban Development smoke-free rule. First, there was support for the rule by the residents, including those that smoke. This provides evidence that similar non-smoking policies will be accepted by residents and possibly desired as motivation to quit. Second, there was a lack of knowledge about the rule among residents, with several residents unaware after its implementation. One resident, when asked when they heard about the rule, replied: “No, they send letters, and then we had to—they sent us a letter and we had to take it to the office. We had to take it to the office, so they’d know that we got the letter. So, and, um, we agreed for this.” Thus, communication prior to and throughout the implementation process is crucial for understanding and, possibly, acceptance. Third, a key component of the implementation of the non-smoking rule is access to cessation support to help residents quit smoking. There was a desire for the materials, and residents responded positively when they were provided. Conversely, one resident who was disappointed about the cessation support indicated that they were not given materials but wished that they had been given materials. Fourth, both administrators and residents were confused about the enforcement of the rule, which should be clarified in the future. This will help inform everybody involved and increase transparency about the entire process. Both were unsure about when enforcement would begin, who would enforce the rule, and what would happen to violators. Lastly, residents had several concerns about the rule, with safety for themselves and their families being the largest concern. Therefore, safety must be a top priority and consideration when implementing the non-smoking rule. Several residents mentioned having to walk to bad neighborhoods to stay compliant or having to leave children inside while they go outside to smoke. The previous points represent several key topics that were learned through the study which were not known before.

### 4.6. Limitations

Due to the iterative nature of qualitative data analysis, these themes are not finite and may evolve as more data is analyzed to ensure dependability, credibility, and transferability over time [39]. Additional focus groups and interviews are planned for the continuation of this study. However, the qualitative design and framework provide depth and a detailed perspective at any point in time. Further, the novel application of the socio-ecological framework to understand a complex issue on multiple levels provides broad perspectives of residents and administrators. Another potential limitation is response bias or self-consciousness. This is inherent to qualitative research, particularly when evaluating a sensitive topic.

Another potential limitation is generalizability, as this study focuses on urban public housing in Washington, D.C. with a predominantly black population and excluding Section 8 housing. This is in contrast to other public housing settings, which may be more rural, have fewer black residents, or differ in percentages of sex.

Another limitation is the scope of this study. The focus was on residents and property administrators, as the most-impacted groups by the rule. None of the upper-level administrators were interviewed, which could result in a less-holistic view of the perceptions of the rule.

### 4.7. Future Direction

Smoking behaviors and rule compliance must be understood in a range of contexts. Considering the recent coronavirus pandemic (COVID-19), it is necessary to explore what effect it has had on residents’ smoking habits, where they smoke, their health, and recidivism if they stopped smoking. Given the impact of COVID-19 on so many aspects of everyday life, an effect on the non-smoking rule and its implementation is likely. Similarly, residents’ marijuana use needs to be better understood as it relates to rule compliance, particularly as it continues to become legalized for both medicinal and recreational purposes. Furthermore, the usage of electronic cigarettes and heated tobacco products by public housing residents deserves additional investigation.

Going a step beyond rule compliance, it will be important to examine the optimal ways to provide cessation support for those who want to quit smoking. Non-smoking interventions that provide cessation support for residents in urban public housing should be implemented in the future. Many residents in this study mentioned the desire to quit smoking and the lack of resources that support it. By providing cessation support to those who want it, there may be greater adoption of non-smoking policies in other scenarios.

Additionally, future research and interventions should be tailored to the specific type of housing residents are living in. In this setting, data was collected from both family and senior/disabled housing units. This distinction should be considered in future research, given the differences and implications of secondhand smoking in each setting.

## 5. Conclusions

A majority of administrators and residents reported positive implications of the mandated Department of Housing and Urban Development rule as means to improve the residents’ health. Implementation processes, however, can be improved, including increased rule communication and signage, clarification of enforcement roles and consequences, maintenance, and safety concerns. Moreover, as the rule motivates quitting attempts, smokers must have access to cessation support and intervention. With input from the communities the rule intends to serve, improvements can be uniquely tailored to different levels of the socio-ecological framework, including individual, interpersonal, organizational, community, and public policy, to maximize impact.

## Figures and Tables

**Figure 1 ijerph-18-08908-f001:**
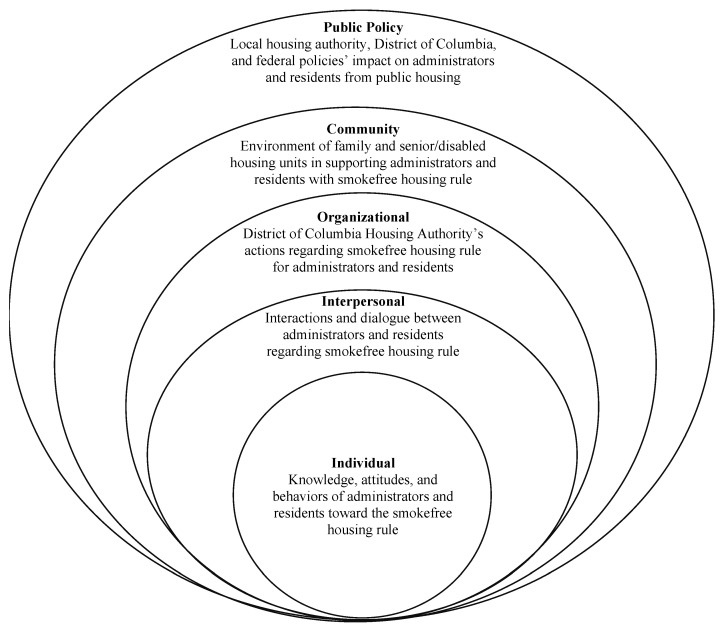
Illustration of the socio-ecological framework specific to the current study.

**Table 1 ijerph-18-08908-t001:** Department of Housing and Urban Development smoke-free rule components and implementation suggestions.

Rule Components	Implementation Suggestions
Ban the use of tobacco products: Waterpipes (hookahs), cigarettes, cigars, and pipes in all outdoor areas up to 25 feet from the public housing and administrative office buildings.	Advertise as smoke-free buildings. Include signage and communication to remind existing tenants, guests, and maintenance workers.
All public housing other than dwelling units in mixed-finance buildings.	The Department of Housing and Urban Development may use the periodic inspections and units to help monitor and confirm whether the policy is being enforced and will take whatever action it seems necessary and appropriate in case of violation.
Public housing authorities may further restrict residents with repeated violations.	Offer cessation support (voluntarily). The Department of Housing and Urban Development encourages public housing authorities to use a graduated enforcement approach that includes written warnings for repeated policy violations before pursuing lease termination or eviction.
Public housing authorities are required to document their smoke-free policies in their plans and require resident engagement and public meetings.	Inform tenants and communicate widely: *Articles in the tenant newsletter, meetings, sending letters, supplying information on the harmful effects of secondhand smoke.*
The use of prohibited tobacco products must be included in a tenant’s lease. Amendments process. Tenants renew their lease annually.	Modify leases. Lease violations: *Residents are responsible for the actions of their household, their guests, and visitors.* *Failures constitute a material and non-compliance with lease agreement. Responsible for all costs to remove smoke odor or residue upon any violation.* *Residents can be charged for property damage that is beyond normal wear and tear, in accordance with 24 CFR 966.4(b) (2). Violation of the smoke-free policy will constitute a lease violation.* *The Department of Housing and Urban Development has not included enforcement provisions in this rulemaking because lease enforcement policies are typically at the discretion of public housing authorities.*

**Table 2 ijerph-18-08908-t002:** Description of focus groups by smoking status.

Focus Group Number	Smoking Status	Total Participants
1	Nonsmoker	11
2	Smoker	7
3	Smoker	8
4	Nonsmoker	11
5	Nonsmoker	3
6	Smoker	9
7	Smoker	6
8	Nonsmoker	6
9	Smoker	8
Total		69

**Table 3 ijerph-18-08908-t003:** Main study themes, meaning units, categories, and code definitions.

Theme	Categories	Definition
Cessation support (*n* = 100 meaning units)	Quit smoking services	Describes cessation services provided to or requested by residents when the rule went into effect.
Communication (*n* = 150 meaning units)	Admin rule communication Resident rule communication	How and when administrators and residents heard about the smoke-free rule, including roles and any information for the tenants.
Enforcement (*n* = 69 meaning units)	Policy execution Reporting violations Eviction	Describes enforcement activities that make sure that residents comply with the rule. May include barriers and facilitators for enforcement as any mention of a resident informing an admin about another resident violating the non-smoking policy.
Lease (*n* = 38 meaning units)	Contract agreement Eviction	Any mention of residents’ leases, including signing an agreement to continue living there and eviction.
Resident engagement (*n* 185 meaning units)	Admin role Admin rule communication Cessation support Resident rule communication Training–guidance	Activities to involve tenants in supporting and adhering to the smoke-free rule.
Signage (*n* = 414 meaning units)	Rule components Signs Smoking place	Signs to promote District of Columbia Housing Authority units as smoke-free. Includes communication to remind existing tenants, guests, and maintenance workers of the smoke-free rule.
Violations (*n* = 568 meaning units)	Compliance/rule-breaking Concerns Consequences Disability Implementation Lease Safety Smoking place Visitors Weather/seasonality	Failures to follow the smoke-free rule, including non-compliance with the lease agreement.

**Table 4 ijerph-18-08908-t004:** Illustrative quotes of major themes at each level of socio-ecological model.

Socio-Ecological Model Levels	Theme	Illustrative Quote		
		Administrator	Resident Smoker	Resident Non-Smoker
Individual	Lease	“The fact that you sign the new lease and— that little fine print that you didn’t read, is in there. You see, and that’s what’s gon’ get you. ‘Cause you know it’s a no smokin…”	“And then we had to go to the rental office and sign a paper that we knew about the rule was goin’ into effect and we’re going to improve our health all.”	“A couple of months ago, they sent us a letter through the mail saying we had to go sign a new lease stating there’s no smokin’ in your unit or on the property, which is good because of health reasons to everyone.”
	Resident Engagement	“And residents had to sign an addendum saying that they’ve been, met with that information. It is the only way that people will follow the rule. And then also, on our part, uh, we were, uh, instructed to do some smoke-free workshops, uh, with them.”	“I’ve heard that there was going to be classes to quit smoking, but people really don’t wanna just walk over there and just be—who wants to be in a class to stop smoking? That’s your own choice, besides we have not been given any support yet.”	“I think that people who smoke were given resources to help them quit smoking, but they don’t wanna quit. They may not smoke outside, but they’ll smoke in their apartment, and it’ll stay in.”
	Violations	“If for instance someone is seen smoking where it’s not designated or not allowed, it hasn’t been made very clear the consequence, but there would be a couple of warnings, and then there would be a 30-day notice, initiated.”	“Now I really don’t know where to smoke. I don’t go outside on the property no more. And then I can’t do too much walking to many places like that, so I might take a puff or two and stuff like that in the halls. But, you know, that’s it because I don’t want problems.”	“I reported a violation, and management say that I needed to write a letter and they were gonna report to the attorney. Then I guess they probably get a warning on what they will do the next time.”
Interpersonal	Communication	“I don’t even know what are the consequences for the person who’s caught smoking might be or if it is said in the policy we received. I think it is a good strategy to decrease second hand smoke for all. But, I’m not sure who is responsible to enforce it.”	“We heard about the rule over the summer, we weren’t prepared for it. Because, number one, they just came out with one letter from HUD that came to us. Probably, we got a letter about—it’s just before August. Then they postponed it a month before they put it in effect.”	“I think that, for the time frame, it was a couple of months. But, it could’ve been, announced or managed better. Like ‘Okay, guys, you know, we’ll have this amount of time, and then maybe another reminder.’ As opposed to, ‘This went into effect—’ it’s just like, ‘You just have to know on your own, but this is what it is.’”
	Enforcement	“Overall, the biggest challenge for actually enforcing it is to identify, unless you know a person is doing it in a unit, but I can’t pinpoint where the smell is coming from because it travels through the air vents. So how do you tell that that’s coming from a particular unit? So, it’s hard to enforce something that- it’s like trying to catch air. Even then, you have to prove it, you know.”	“If I knew somebody was smoking in a place where they’re not supposed to be, I wouldn’t do anything about it. I would mind my business, ‘cause it ain’t none of my business.”	If I see somebody smoking where they’re not supposed to be smoking, I really don’t know what I would do. I probably would tell them, ‘You are not in the right place to be smokin’ and contributing to air pollution.’”
	Lease	“It’s supposed to be three warnings. On that third warning, that’s what they’re supposed to do, somethin’ about it but the rules are not being followed. The rules are not being followed neither by HUD, neither by the District of Columbia Housing Authority, nor by the residents.”	“I signed mine when I paid my rent. They said, ‘Before we—accept your rent, you have to sign this document right here.”	“A couple of months ago, they sent us a letter through the mail saying we had to go sign a new lease—stating there’s no smokin’ in your unit or on the property... We heard talk about it in a couple of the meetings that they had, that they was gonna stop everybody from smokin’ on the property…”
Organizational	Cessation Support	“I’ve got good responses from folk who said that they wanna quit. They want more information. But what we lacking is some of the products that help you to stop smoking, it was gonna be available, but we haven’t connected with a source to have that.”	“Yes (cessation services were provided). There was a lot of them. They even was giving people the number to get some free patches from some—free cigarette no-smoking patches and everything.”	“I don’t recall gettin’ no type of resources other than sayin’ that you could smoke in public places—not public but outside places like at a park or somethin’ like that…”
	Enforcement	“We don’t enforce it at all. That’s not our role at all, to enforce. Enforcement side is the management side.”	“Well, they said they gonna put these smoke things in our apartment, monitors or whatever they were.”	“No, I would not (report it to the building management)... I would just remind them that it’s in effect, but I wouldn’t report it to the management, no.”
	Signage	“There are signs indicating that the District of Columbia Housing Authority is a smoke-free area, they came from the District of Columbia Housing Authority. There are not enough signs, because some places should have the signs and there are no signs out in areas like the courtyards because kids are up there.”	“There are reminders of the rule around the property, they finally put signs up, *No Smoking*, but they still smoke.”	“There’s no way to tell, like, where you can smoke and where you can’t smoke. You know, they have signs that say (you can’t smoke) within 25 feet, and then they have another sign that say within 50 feet.”
	Violations	“They said, ‘No, we’re not gonna do that (put detectors in unit).’ So how can you enforce it? And they said, well kinda like we really can’t enforce it unless other people enforce it for us. So, you’re gonna have to have someone tell on you, on the smoker, or if maintenance goes into your unit and see an ash tray or a pack of cigarettes.”	“I’m just sayin’ it’s certain management and certain security patrols wanna make their own rules up where you should be. It is not clear what happens if someone doesn’t follow the rule.”	“What happens if someone is, uh, smoking, where it’s not allowed? Nothing. They just ask ‘em—when there was security officer could tell them that they couldn’t smoke on the property, and they would go across the street.”
Community	Signage	“And one in particular had built a gazebo. They had ashtrays that were mounted, and then they end up smoking outside the property line because of the smoke-free area signs. They end up smoking across the street, which caused a problem in the community.”	“Around the property there aren’t signs that reminds about the no smoking, but there are when you’re inside, but when you’re outside the fence, they don’t have no signs out there.”	“They finally put signs up, “No Smoking,” but they still smoke. I can see how it is when it’s winter and it’s cold out there, some people don’t like to go outside standin’ out in that cold to smoke a cigarette.”
	Violations	“You know, they were told about it, but, they smoked all in their place and in here all the way up to the time they say they couldn’t smokin’ because it is too cold, or dangerous, or even some for mobility issues and then some they started goin’ cross the street, which caused a problem in the community.”	“You don’t wanna be walkin’ out there every 15 min. The weather don’t permit. It’s cold. It’s windy, and it—people that livin’ across the street look at you like you’re hoodlums or bums or whatever. It’s dangerous.”	“I don’t like it. I mean, why can’t she smoke in her own home? because she has a young kid, and it’s not safe to leave her alone. I will let her smoke on the balcony.”
Public policy	Lease	“I‘ll probably tell you, ‘And you get one more we’re gonna evict you.’”	“Before the rule went into effect, we all got the last notice that the rule was goin’ into effect. And then we had to go to the rental office and sign a paper that we knew about the rule was goin’ into effect.”	“I think the D.C. Housing Authority included the smoke-free rule into the leases because they want everyone to be safe in their homes, and they don’t want to increase any health problem within D.C. Housing.”
	Violations	“But housin’, um, even when they, I don’t know if they gonna let us know or whatever. But it’s gonna be a thing whereas they’re gonna have to give you warnings, and after so many warnings I have housin’ puttin’ you out.”	“They just put the rule down, but they didn’t enforce it. They just put the rules out there, but they did nothing about the rules.”	“It’s just, when they have a designated-smoking area, they had benches where they could sit down at. It’s a gazebo, and the ashtray, designated ashtray, there that the public housing put up. But then when the law came, they say now they cannot use ‘em. It doesn’t make good sense.”

## Data Availability

The data used to support the findings of this study are available from the corresponding author upon request.

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
