# Peer review of "Implementation of the Department of Housing and Urban Development’s Smoke-Free Rule: A Socio-Ecological Qualitative Assessment of Administrator and Resident Perceptions"

_ijerph, 2021, doi:10.3390/ijerph18178908_

Round 1

Reviewer 1 Report

This is an interesting, well-written article concerning an important social problem. I highly appreciate the tables presenting the results. However, the Methods section should be improved. 

In study sample section (methods 2.1.) there is no clear information about the precise number of participants in this study.  7 administrators participated in the interviews and 21 residents participated in both focus groups and interviews, therefore the whole number of participants of this study and people who participated only in interviews or focus group is missing. I suggest to begin with this information and then present details about participants. In demographic information section (methods 2.2.1.) there is information that „not all participants completed the survey”, however the number of these participants is also required. In focus groups section (methods 2.2.2.) there is information that „the number of participants in each focus group ranged from 6 to 15”, this also should be corrected to a precise number of participants in each focus group. 

The discussion section 6.Future direction could also include information about usage of electronic cigarettes and heated tobacco, which popularity is constantly increasing and should be included in future studies. 

Some spelling mistakes should also be corrected.

Author Response

In study sample section (methods 2.1.) there is no clear information about the precise number of participants in this study.  7 administrators participated in the interviews and 21 residents participated in both focus groups and interviews, therefore the whole number of participants of this study and people who participated only in interviews or focus group is missing. I suggest beginning with this information and then present details about participants.

Reply: Section 2.1 was updated in the beginning to indicate that 109 residents and 7 administrators were the total number of participants. This number was broken down to indicate 69 residents participated in 9 focus groups, 61 residents participated in in-depth interviews, and 21 of the focus group participants were also in-depth interview participants.

In demographic information section (methods 2.2.1.) there is information that, “not all participants completed the survey”, however the number of these participants is also required.

Reply: The number of focus group and in-depth interview participants with missing data was added to the section, indicating 21 focus group participants and 4 interview participants.

In focus groups section (methods 2.2.2.) there is information that the number of participants in each focus group “ranged from 6 to 15”, this also should be corrected to a precise number of participants in each focus group.

Reply: A table (Table 2) was added which lists each focus group number, whether or not the group was non-smoker or smoker, and the total number of participants in each group.

The discussion section 6. Future direction could also include information about usage of electronic cigarettes and heated tobacco, which popularity is constantly increasing and should be included in future studies.

Reply: We agree with your comment and added a sentence indicating electronic cigarettes and heated tobacco as a direction for future studies.

Some spelling mistakes should also be corrected.

Reply: Spell-check, Grammarly, and a detailed perusal were used to fix spelling mistakes.

Reviewer 2 Report

Overall, this paper was a good overview of an assessment on the Mandatory Smoke-Free Rule in public housing implemented by the U.S Department of Housing and Urban Development (HUD) in July 2018. The study uses sound qualitative methods and encompasses different levels within the socio-ecological framework.

Minor suggestions, mostly in spacing and structuring have been suggested:

Under Section 2.2.1

- Correct to: Resident demographics were obtained.

- Move “Additionally, snowball sampling was conducted, in which focus group participants could bring friends to participate in the focus group.” to methodology section.

Under section 3.1

-Correct spacing in: “The mean age was 56.37, with a standard deviation of 12.8.”

Under section 4.1

-Correct spacing before the following sentence: “Past research suggests that most of the support is driven by perceived health benefits of the rule [30].”

Author Response

Overall, this paper was a good overview of an assessment on the Mandatory Smoke-Free Rule in public housing implemented by the U.S. Department of Housing and Urban Development (HUD) in July 2018. The study uses sound qualitative methods and encompasses different levels within the socio-ecological framework.

Minor suggestions, mostly in spacing and structuring have been suggested:

Under Section 2.2.1

- Correct to: Resident demographics were obtained.

Reply: This change was made.

- Move “Additionally, snowball sampling was conducted, in which focus group participants could bring friends to participate in the focus group.”

Reply: This sentence was moved to the “2.1 Study sample” section under the “2. Methods” section and clarified that it wasn’t truly snowball sampling.

Under section 3.1

-Correct spacing in: “The mean age was 56.37, with a standard deviation of 12.8.”

Reply: This spacing was corrected.

Under section 4.1

-Correct spacing before the following sentence: “Past research suggests that most of the support is driven by perceived health benefits of the rule [30].”

Reply: This spacing was corrected.